# Replication Study of Molded Micro-Textured Samples Made of Ultra-High Molecular Weight Polyethylene for Medical Applications

**DOI:** 10.3390/mi14030523

**Published:** 2023-02-24

**Authors:** Francesco Modica, Vito Basile, Rossella Surace, Irene Fassi

**Affiliations:** 1CNR-STIIMA Institute of Intelligent Industrial Technologies and Systems for Advanced Manufacturing, Via P. Lembo, 38F, 70124 Bari, Italy; 2CNR-STIIMA Institute of Intelligent Industrial Technologies and Systems for Advanced Manufacturing, Via A. Corti, 12, 20133 Milano, Italy

**Keywords:** micro-EDM structured surface, micro-injection molding, micro-texturing, Ultra-High Molecular Weight Polyethylene

## Abstract

In articular joint implants, polymeric inserts are usually exploited for on-contact sliding surfaces to guarantee low friction and wear, a high load-bearing capacity, impact strength and stiffness, and biocompatibility. Surface micro-structuring can drastically reduce friction and wear by promoting hydrostatic friction due to synovial fluid. Ultra-High Molecular Weight Polyethylene (UHMWPE) is a suitable material for these applications due to its strong chemical resistance, excellent resistance to stress, cracking, abrasion, and wear, and self-lubricating property. However, surface micro-texturing of UHMWPE is hardly achievable with the currently available processes. The present study investigates UHMWPE’s micro-textured surface replication capability via injection molding, comparing the results with the more easily processable High-Density Polyethylene (HDPE). Four different micro-texture cavities were designed and fabricated on a steel mold by micro-EDM milling, and used for the experimental campaign. Complete samples were fabricated with both materials. Then, the mold and samples were geometrically characterized, considering the dimensions of the features and the texture layout. The replication analysis showed that HDPE samples present geometrical errors that span from 1% to 9% resulting in an average error of 4.3%. In comparison, the UHMWPE samples display a higher variability, although still acceptable, with percentage errors ranging from 2% to 31% and an average error of 11.4%.

## 1. Introduction

Arthroplasty is a surgical procedure used to restore the functionality of a joint when medical treatments are no longer effective. It often requires the use of an artificial implant designed to reproduce the articular kinematics and dynamic behavior. The research in the field is very active, focusing mostly on the development of materials and related manufacturing processes to improve osseointegration, increase the biocompatibility and wear resistance in order to reduce the time of hospitalization and the risk of infections, and extend the device lifetime and optimize its overall performance, to reduce the risks of secondary arthroplasty.

In current artificial joint implants, titanium alloys are typically employed for the structural components that require load bearing and integration in bones. Titanium alloys are preferred as bearing components due to their mechanical performance, biocompatibility and corrosion resistance, but their debris dispersion can trigger osteolysis [1,2].

At the same time, polymeric inserts are generally used for at least one of the coupling interfaces, such as the radio-capitellum in elbow implants, the patella in knee implants, the acetabular cup in hip joint implants, and the humerosocket liner in shoulder implants. Polymeric materials are preferred over ceramics or metals due to their mechanical and tribological properties, which are closer to the ones between natural surfaces. Moreover, their lower dynamic friction coefficient reduces the risk of debris generation, thus reducing the risk of infections and biological reactions such as inflammation, bone resorption, or periprosthetic osteolysis [1]. Indeed, the incidence of osteolysis increases as the rate of wear increases. For example, the current service life of an artificial hip joint prosthesis is about 15 years and a research gap must be filled to improve the quality and life of these products. Improvements are required in the materials’ performance, biomechanics, biotribology, biomaterials, mechanical analysis methodology, and physiology [3].

The polymeric components in artificial joints act as cartilage surfaces in natural joints characterized by a low coefficient of friction (CoF), about 0.02 [1]. They guarantee biocompatibility, low friction and wear, a high load-bearing capacity, stiffness, and damping of impact loads.

Among biomaterials, Ultra-High Molecular Weight Polyethylene (UHMWPE) has gained widespread application due to its superior properties [4,5] compared with other materials i.e., Polyethylene (PE), Polypropylene (PP), Polyether Ether Ketone (PEEK), Polyamide PA66, zirconia-on-PE, alumina ceramic [1,4]. The current state-of-the-art on materials for hip prostheses has been widely presented in a recent review paper [6]. Two procedures are adopted in hip joint surgery: total hip arthroplasty and hip resurfacing [3]. The former consists of replacing the femur head, femoral stem, and acetabulum cup, while the latter consists of the replacement of the bearing couple (femoral head and acetabular cup) at the interface between the femoral stem. In hip resurfacing, it is not required to replace the femoral stem with an implant component (Figure 1).

The bearing couple is the main functional component for load-bearing and movement articulation in all joint implants since it provides continuous contact and mechanical action transmission between structural components [3]. Bearing couples are subjected to friction, wear, and surface damage, thus affecting the overall performance and the total duration of the prosthesis. In the last decade, several studies have focused on the bearing couple in order to increase the life of implants and minimize implant failures [3].

## 2. Surface Micro-Texturing for Enhanced Tribological Properties in Bearing Couples

In order to extend the working life of the implants, a successful approach requires reproducing not only the cartilage properties but also the natural hydrostatic friction. This last feature is given by the synovial fluid, which works as a lubricant film at the cartilage–bone interface [7], thus guaranteeing low friction and drastically reduced wear. UHMWPE is an engineering thermoplastic semicrystalline polymer composed of two interpenetrating phases: an ordered crystalline phase and an amorphous disordered phase [8]. As specified by ASTM [9], it has an exceptionally high molecular weight, greater than 3.1 million atomic mass units (AMUs). This very high molecular weight facilitates high entanglement density in its amorphous phase and superior toughness compared to other homopolymers. The advantages of UHMWPE include excellent stress resistance, high impact strength, excellent resistance to cracking, abrasion, and wear, a strong chemical stability, excellent dielectric properties, and self-lubrication. The U.S. Food and Drug Administration (FDA) approved it for use in the food, beverage, medical, and pharmaceutical industries.

Moreover, in UHMWPE implants, osteolysis is infrequent when the wear rate is up to 0.1 mm/y, and it is almost absent when its value is 0.05 mm/y or below [1]. A wear rate of 0.1 mm/y will result in a CoF threshold of about 0.02, which can be achieved with hydrodynamic friction [10].

In order to improve the tribological properties and promote hydrodynamic friction through the lubricant film between interface components, surface micro-texturing can be exploited.

Usami H. et al. [10] proposed surface micro-texturing on samples made of aluminum alloy for casting (AC8A-T6). Arrays of micro-dimples with diameters in the range ∅200–300 μm, depths of 5–10 μm, and area fractions of 10 and 40% were manufactured by micro-tilling (interrupted milling) on disk samples. Under lubricated conditions, the tribological properties were evaluated using a ring-on-disc-type testing apparatus. Experimental results show a stable CoF reduction, up to −35%, with the micro-textured surfaces. A dependence of the CoF on the area fraction was also found: lower CoF (about 0.01) was achieved with an area fraction of 10%, thus proving that micro-feature density is an important design parameter. Pratap et al. [11] studied the effects of surface micro-texturing on Ti-6Al-4V alloy samples used in the hip joint prosthesis. Flat, hemispherical, and conical micro-dimples with a diameter of 200 μm were realized by micro-milling and surfaces were characterized by measuring roughness, micro-hardness, surface wettability, and CoF. Results proved that surface micro-texturing enhances the hydrophilicity of the sample. Hemispherical dimples allow higher tribological properties than other geometries (flat and conical). Surface micro-texturing results in a CoF reduction of about 25–27% under different load pressure conditions [10]. Dong Y. et al. [12] studied the tribology of micro-texturing and low-temperature plasma carburizing on CoCrMo structural components sliding on UHMWPE interface components in a hip joint prosthesis. Micro-features were hemispherical with diamond micro-dimples having widths of 55–190 μm and depths of 1.5–9.5 μm. Hexagonal arrays with inter-axes of 80 and 500 μm for the hemispherical and diamond features, respectively, were manufactured by mechanical indentation on the CoCrMo hip head surfaces. Experimental tests with a hip simulator and interposing a lubrication film between the sliding components revealed a CoF reduction of 20%. Jamari J. et al. [13] implemented a numerical model to evaluate the micro-dimples’ effects on the surface of interface components in the hip joint prosthesis. Contact pressure and wear were significantly reduced by surface micro-texturing, up to 24.3% and 31% for linear and volume wear, respectively. A recent review over 30 years showed that the computational simulation is an effective approach in assessing bearing, friction, wear, surface damages, performance, and failure in artificial joints [3].

This analysis suggests that suitable surface texturing on the UHMWPE interface components of joint implants [14,15] will improve the tribological properties. However, two crucial issues should be addressed: the UHMWPE processing technology and its replication capability of textured surfaces at the micro-scale.

UHMWPE has an essentially zero melt flow index, poor dissolution, and extremely high melt viscosity, and it is hard to process by conventional extrusion and injection molding (IM) technologies [16,17,18,19,20]. Indeed, the main processing technology for UHMWPE is compression molding, which, however, has a lower replication capability with respect to injection molding and it is not suitable for mass production. In other words, the injection molding of UHMWPE is still a challenge, especially with the need for micro-features to be replicated accurately and repeatability [17,20]. Yilmaz G. et al. [17] addressed the issue of the delamination on the injection-molded UHMWPE parts. The drawback was significantly reduced through mold thermal insulation or by introducing supercritical nitrogen or carbon dioxide into the polymer melt, which decreases UHMWPE viscosity and injection pressure, thus reducing the risk of degradation [21]. Heidari B.S. et al. [18] combined information from process simulations and experimental tests aimed at the definition of optimized injection molding process parameters. Sánchez-Sánchez X. et al. [19] implemented an ultrasonic-assisted injection molding to optimize the filling phase of UHMWPE. Recent progress was achieved with the high-temperature melting (HTM) technique using a significantly higher melt temperature and injection pressure to process UHMWPE by injection molding [20]. These issues are amplified with surface micro-texturing, where the replication capability of molds is crucial.

In this work, the processability of UHMWPE via micro-injection molding and the micro-texturing surface replication capability are investigated, following the research workflow shown in Figure 2. First, four different micro-textures were designed in terms of texture geometry and pattern. Then, a mold insert was prepared, with the selected micro-textures fabricated via micro-EDM milling. The micro-textures were characterized by confocal microscopy. UHMWPE samples were then produced by micro-injection molding, and process parameters were adjusted to improve the replicability, pushing the limits of the machine. Then, the molded parts were characterized to assess the overall process replication capability.

## 3. Micro-Textures Design

In order to improve the tribological performance of artificial joint coupling surfaces, several characteristics are required: a stable micro-film of synovial fluid to enable hydrodynamic lubrication; reservoirs of fluid on the components’ interface; micro-traps for collecting undesired micro-debris; a low CoF; and adequate surface wettability to promote lubrication.

These criteria result in a set of design specifications for the micro-textures that should exhibit [22]:−a pattern of micro-cavities;a symmetrical design to avoid anisotropy;a cavity area fraction (AF) in the range 10–40%, where AF is defined as the ratio of the cavity area and pattern unit cell area [10].

Surface patterned micro-textures result from a combination of three main characteristics: micro-features of the texture (cavity, pillars, etc.); the pattern; and the hierarchy of patterns (Figure 3). Combining these elements, four micro-textures were conceived (Figure 4) and the mold inserts were consequently designed, as shown in Figure 5. The proposed micro-textures were chosen following the above-mentioned guidelines and the following considerations. Hemispherical micro-cavities (dimples) were chosen according to the results of previous studies [11], where this shape resulted in improved tribological properties compared to other shapes (flat or conical). The micro-cavities’ dimensions were varied to study the effects of the AF parameter, ranging from 10% to 40%. Symmetrical design specification suggests a hexagonal layout of micro-cavities. Finally, a texturing hierarchy was defined in order to improve the uniform distribution of the synovial fluid (lubricant film) at the interface, increase the Afs, and maintain the symmetry of the layouts.

All micro-textures are based on spherical shapes, which are domes on the mold and cups on the samples. HEX300 and HEX200 are hexagonal patterns of cavities with diameters of 300 and 200 µm, respectively. The HEX300-150 is designed with a hierarchy of two hexagonal patterns of cavities with diameters of 300 and 150 µm. HEX300 with channels (HEX300+CH) is a hierarchized micro-texture obtained from HEX300, adding a pattern of 50 µm width channels between cavities. In all textures, the pattern unit cell has a width of *W* = 0.700 mm and a length of *L* = 1.212 mm (Figure 4). The micro-cavities’ depth is *H* = 20 µm. Parameters are reported in Table 1.

## 4. Materials and Methods

A special grade of UHMWPE, GUR^®^ 5113 (Celanese, Lanaken, Belgium), which the manufacturer recently introduced as suitable for the injection molding process, was used during the tests. In order to assess the injection molding process capability with this material, it was compared with an easier moldable material such as a High-Density Polyethylene (HDPE). HDPE has a high strength-to-density ratio, whereas UHMWPE contains long polymer chains with very high molecular weights. A Versalis Eraclene^®^ MP90C HDPE (Versalis S.p.A., San Donato Milanese, Italy) was chosen for the comparison. As a reference, Table 2 lists the main properties of the GUR^®^5113 material compared to a generic UHMWPE [23] and to HDPE.

The manufacturing process chain is depicted in Figure 6.

The mold micro-textures were fabricated via micro-EDM milling using a micro-EDM machine (Sarix µ-EDM SX200) characterized by three cartesian axes having a resolution of 0.1 µm and repeatability of ±2 µm. The workpiece material was a mold steel, while two standard-size tool electrodes were adopted for the micro-milling: a cylindrical rod with a diameter of 0.4 mm for machining the cavities HEX300 and HEX200, and a cylindrical rod with a diameter of 0.15 mm for machining the cavities HEX300-150 and HEX300+CH. The tool electrode material was tungsten carbide (WC), and the dielectric fluid was a hydrocarbon oil. The process parameters reported in Table 3 were chosen considering a finishing regime and accordingly with the database supplied by the machine manufacturer and to previous studies [24], considering the combination of WC tool electrode and a mold steel workpiece.

The µ-IM was performed exploiting a DesmaTec FormicaPlast 1K machine (DesmaTec GmbH, Achim, Germany), capable of injection speed up to 500 mm/s, injection pressure up to 300 MPa, and maximum injection volume of 150 mm^3^. In this process, the thermoplastic material is heated, melted, and injected into a closed cavity through a plunger and then subjected to holding pressure. After cooling, molded parts are taken out, and the cycle is repeated.

Profiles and surface roughness of inserts and samples were acquired via confocal microscope Zeiss Axio CSM 700 (Carl Zeiss Microimaging GmbH, Jena, Germany). The acquired images have a spatial resolution Rs of 1.824 μm/pixel that allows for a feature resolution Rf of 5.5 μm considering 3 pixels spanning the minimum size feature and a measurement resolution Rm of 0.1824 μm.

## 5. Characterization of the Mold Surface Micro-Textures

The machined mold inserts, ejection side, and injection side with the textured surfaces are shown in Figure 7. Table 4 reports machining performance in terms of depth error (measured by electric touch in the μ-EDM machine), machining time, Material Removal Rate (MRR, volume removed from the workpiece in the unit of time), and Tool Wear Ratio (TWR, the ratio between the volume of worn electrode tool and the volume eroded from the workpiece) for each micro-textured cavity. In particular, there was a tight agreement in depth error, while the differences in terms of MRR and TWR were related to the electrode size. MRR decreased because, for a smaller tool section, it is necessary to use a longer toolpath to cover a unit of area.

The measurements of the textured cavities and the corresponding standard deviation calculated for each feature are reported in Table 5, while Figure 8 shows their confocal acquisition. The comparison with the nominal dimensions (Figure 4 and Table 5) confirms the good execution of the machining.

## 6. Micro-Injection Molding

Due to the complexity of the µ-IM process, many factors affect the quality of the product. The setup of the process parameters requires accurate adjustments considering cavity geometries and materials. The consolidation of UHMWPE resin particles is a slow process due to the time needed by chains with high molecular weight to entangle with chains in adjacent particles and co-crystallize, assisted by pressure and temperature. Thus, the parameters need to be carefully evaluated. For HDPE, the process parameters were set starting from the literature and previous studies [25,26,27,28] and then modified.

Afterward, the experimental campaign with UHMWPE was carried out. With respect to HDPE, the holding pressure was increased due to the higher shrinkage of UHMWPE; also, the melt and mold temperatures were increased according to the manufacturer datasheet. The injection molding process parameters are summarized in Table 6.

The HDPE samples were scanned (Figure 9) via a confocal microscope and geometrically characterized (Table 7), showing a high replication accuracy.

The obtained parts are shown in Figure 10b (HDPE) and Figure 10c (UHMWPE). The experimental results show that the GUR^®^5113 is able to fill the micro-cavities. Due to the high-value settings of the process parameters, the samples present a small quantity of flash (Figure 10c); however, at the present stage, it can be considered an acceptable drawback for reaching the best mold replication. The process parameters influenced the height and shape of the micro-textures. The injection velocity is a very significant factor, as also confirmed in the literature [29].

During the processing of UHMWPE, it was observed that the flow-out of the molten material is irregular and flaky, likely due to its highly entangled polymer chains structure. Moreover, in some samples, the skin layer appears delaminated, as also referred to by other authors [30].

The delamination layer can be attributed to two main factors. The first is the excessive shear stress, during the filling and holding stages, related to the high viscosity of the material and to the rapid cooling of the skin. The second factor can be attributed to UHMWPE’s intrinsic properties that encourage easy layering due to the high entanglement degree of molecular chains. Separate entangled chain bundles require a longer time and high temperature to join the core during the holding phase [30].

These phenomena, irregular flow, and delamination of UHMWPE require further investigations.

The UHMWPE samples, shown in Figure 11, were geometrically characterized (Table 8), showing good replication accuracy. Regarding the surface roughness, the acquired parameter Sa shows a difference between the mold and parts (0.3 and 0.2 µm). The molding conditions at the polymer–mold interface play a fundamental role during filling, and this result is mainly due to the heat transfer and the slippage between the melt and steel mold.

Figure 12 and Figure 13 show the summary of the percentage errors calculated as the difference between the molded samples and mold dimensions using values of Table 5, Table 7 and Table 8. The formula used is the following:(1)$$E=\frac{\left(D_{m}-D_{s}\right)}{D_{m}}\cdot 100$$
where *D* represents the measure of a generic feature (diameter, depth, distance) of the mold (*m*) and sample (*s*). The most evident result is that the replication of HDPE samples is better than UHMWPE, as expected and already mentioned, due to the high viscosity of the second material. The percentage errors depend on the type of feature. For HDPE samples, the percentage errors span from 1% to 9% resulting in an average error of 4.3%. In the case of UHMWPE, the percentage errors span from 2% to 31%, resulting in an average error of 11.4%. The higher errors occur on the micro-feature depth with a nominal value of 20 μm, in which an error of few units produces a high value of percentage deviation. Conversely, the diameters are well replicated for all samples where the maximum error is under 10%.

Additional qualitative analysis on molded samples was performed by means of a high-resolution optical profilometer Sensofar S Neox (Sensofar group, Barcelona, Spain), set with a confocal acquisition method and magnification of 10× (Figure 14 and Figure 15).

Figure 14 shows the high quality of the HDPE samples. The replication capability is uniform on all surfaces, and no relevant differences are visible from the gate region (left side) to the end of the cavity (right side).

As shown in Figure 15, the molding quality of the UHMWPE samples degrades along the sample length. In particular, the molding quality and replication capability appear acceptable closer to the gate region (left) and for about two-thirds of the sample length, while they degrade approaching the final part of the cavity (right). This result can be explained by considering the high material viscosity and the micro-injection molding parameters. In the filling phase, the cavity regions closer to the gate are characterized by higher injection pressures and melt temperatures. Material viscosity, in conjunction with the cavity geometry, results in a pressure drop and a drastic cooling of the melt front, thus determining a low molding quality. In order to avoid this issue, injection molding process parameters must be optimized by maintaining higher values of injection pressure and temperature along the whole filling phase. The Celanese GUR^®^5113 material was introduced by the manufacturer as an injection molding grade starting from the compression molding grade GUR^®^ UHMWPE, and little experimental information is available. However, more efforts to better characterize the material–process–product relationship are required in order to improve the process capabilities. Thus, the experimental results verified that, differently from previous research, UHMWPE samples can be micro-molded by the injection process. However, the replication capability of micro-features and micro-textures is still a challenging task due to the rheological properties of the material.

It must also be underlined that the mold cavities are, on purpose, narrow and generally difficult to fill. The defects recorded on the samples during the experimentation could not appear with thicker cavities or cavities with higher aspect ratio, as demonstrated by the fact that defects occur far from the gate. The geometry was designed with the goal of studying the limits of injection molding when UHMWPE is injected, providing a helpful indication for prosthesis mold design.

## 7. Conclusions

The present study investigates UHMWPE’s micro-texturing surface replication capability via injection molding. Four different micro-textures cavities were designed and fabricated on a steel mold by micro-EDM milling and used to test the micro-injection process replication. Two polymers, HDPE and UHMWPE, were processed via micro-injection molding, comparing the replication results. The analysis shows that HDPE samples present percentage errors that span from 1% to 9% resulting in an average error of 4.3%. In comparison, the UHMWPE samples display percentage errors ranging from 2% to 31%, with an average error of 11.4%. Higher errors occur on the micro-feature depth with a nominal value of 20 μm, in which an error of a few units produces a high value of percentage deviation. Conversely, the diameters are well replicated for all the samples where the maximum error is under 10%.

The experimental results confirm that UHMWPE samples can be successfully fabricated by micro-injection molding, although the replication capability of micro-features and micro-textures is still challenging due to the rheological properties of the material. High-resolution microscopy reveals a lower quality in the filling of UHMWPE samples compared with the HDPE, thus proving that the process can be further optimized. Further research will regard, as mentioned, the optimization of injection molding process parameters and, in particular, the accurate definition of injection pressure and temperature, whose setting is of paramount importance. In fact, the molding conditions at the polymer–mold interface play a fundamental role. Moreover, a new formulation of a medical-compliant injection moldable Celanese GUR^®^, showing improved viscosity, is under processing and testing. Finally, further research will be focused on the tribology of the developed surface micro-texturing. An experimental campaign will be aimed at accurately measuring the coefficient of friction and wear rates in different operation conditions.

## Figures and Tables

**Figure 1 micromachines-14-00523-f001:**
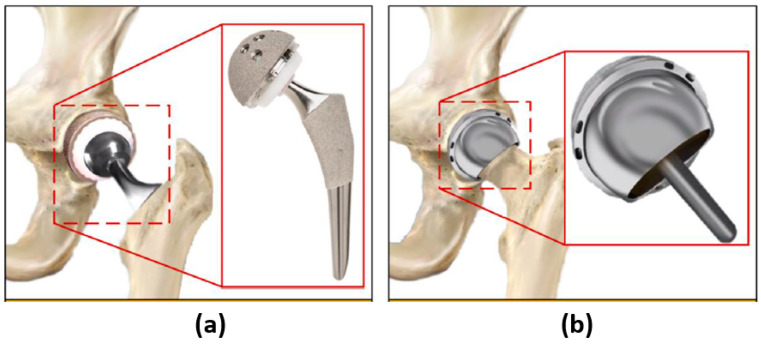
Total hip arthroplasty (**a**) and hip resurfacing (**b**). Reprinted/adapted with permission from Ref. [3]. 2022, Jamari, J.; Ammarullah, M.I.; Santoso, G.; Sugiharto, S.; Supriyono, T.; Permana, M.S.; Winarni, T.I.; van der Heide, E.

**Figure 2 micromachines-14-00523-f002:**
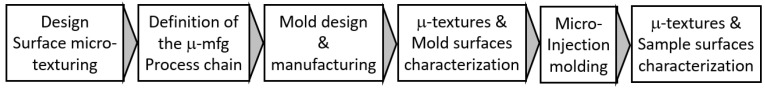
Research workflow.

**Figure 3 micromachines-14-00523-f003:**
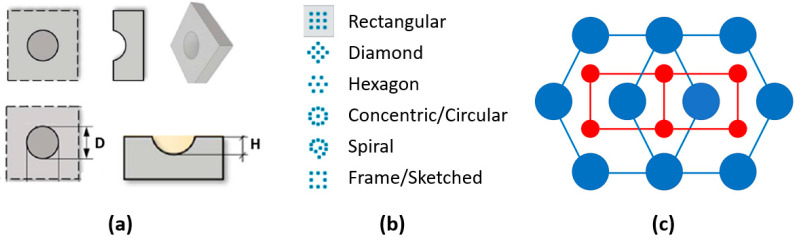
Cup-shaped micro-textures: (**a**) unit feature; (**b**) pattern; (**c**) hierarchy.

**Figure 4 micromachines-14-00523-f004:**
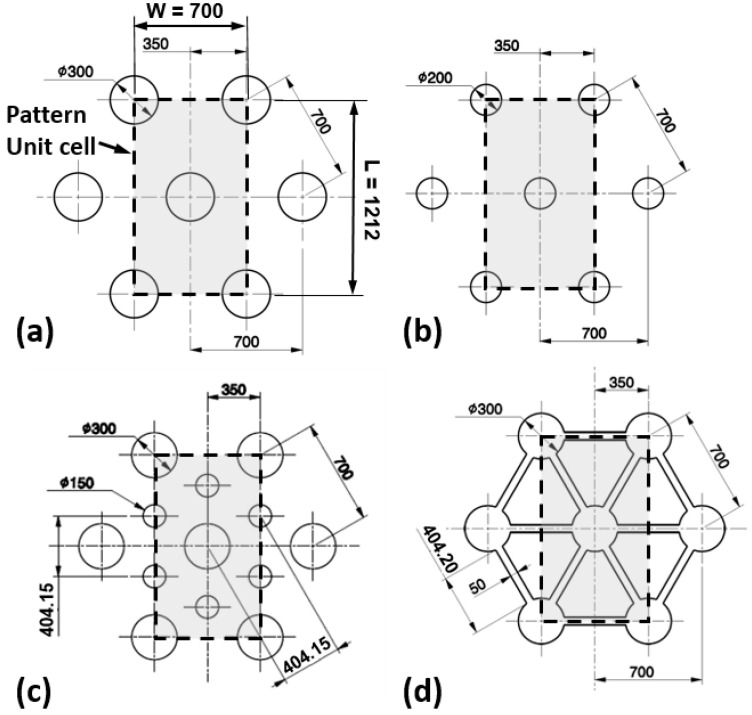
Micro-textures design. (**a**) HEX300; (**b**) HEX200; (**c**) HEX300-150; (**d**) HEX300 with channels (µm).

**Figure 5 micromachines-14-00523-f005:**
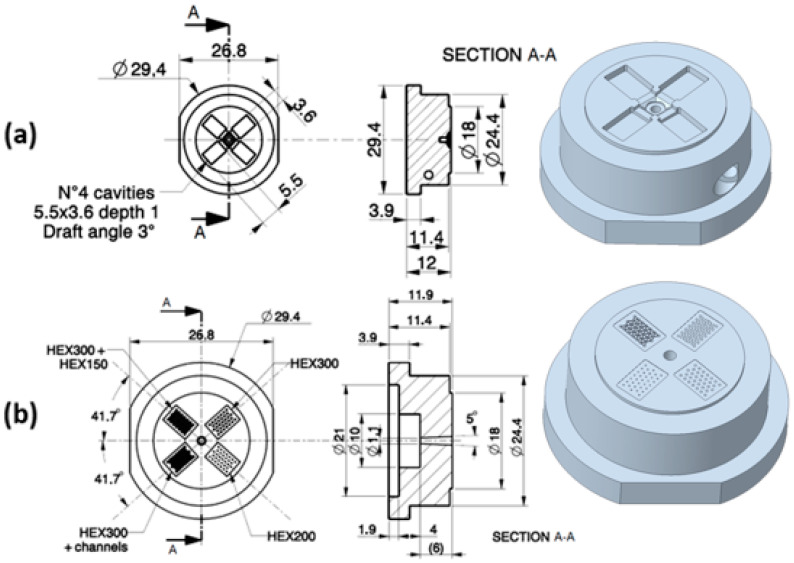
Mold inserts. (**a**) Ejection side; (**b**) injection side with dome micro-textures (mm).

**Figure 6 micromachines-14-00523-f006:**
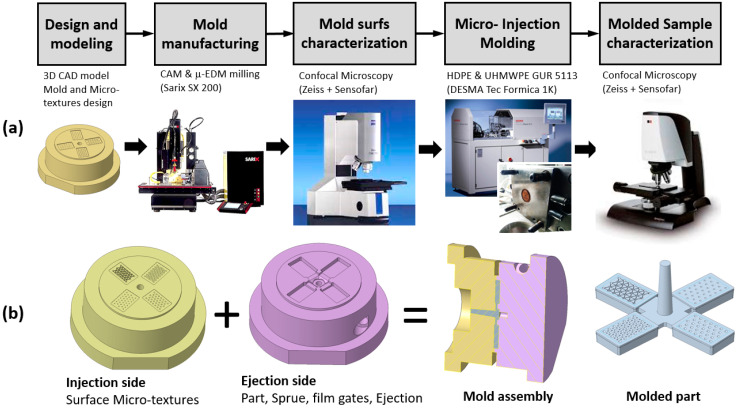
Micro-manufacturing process chain (**a**); mold components and molded samples (**b**).

**Figure 7 micromachines-14-00523-f007:**
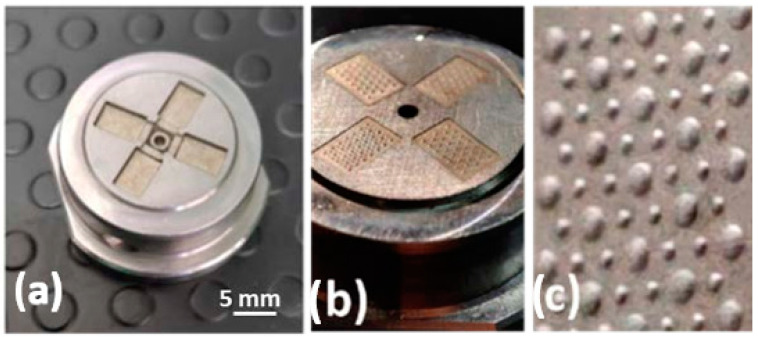
Mold inserts. (**a**) Cavities insert, ejection side; (**b**) micro-textured insert, injection side; (**c**) detail of domes.

**Figure 8 micromachines-14-00523-f008:**
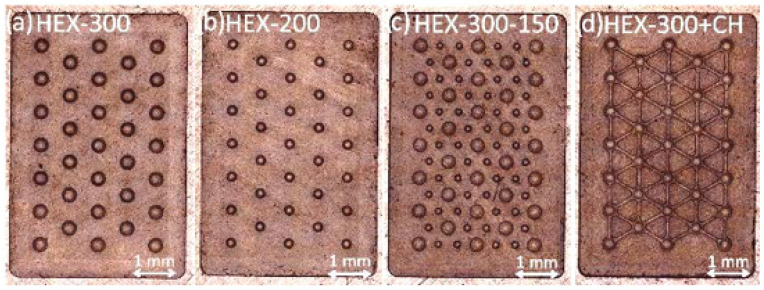
Micro-EDM textured cavities: (**a**) HEX-300; (**b**) HEX-200; (**c**) HEX-300-150; (**d**) HEX-300+CH.

**Figure 9 micromachines-14-00523-f009:**
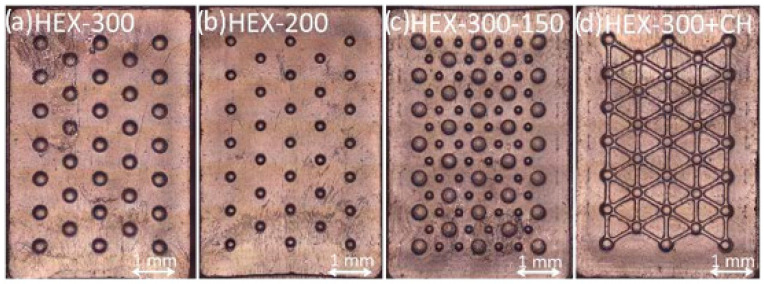
HDPE sample: (**a**) HEX-300; (**b**) HEX-200; (**c**) HEX-300-150; (**d**) HEX-300+CH.

**Figure 10 micromachines-14-00523-f010:**
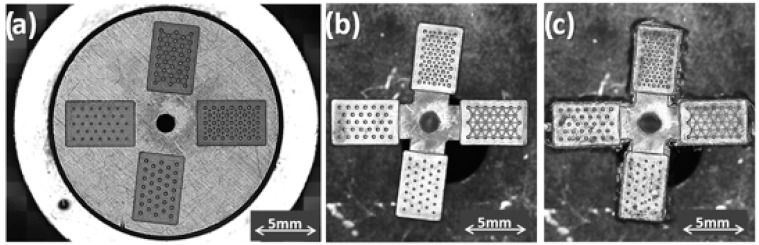
Mold cavities (**a**) and molded samples (**b**) HDPE, (**c**) UHMWPE.

**Figure 11 micromachines-14-00523-f011:**
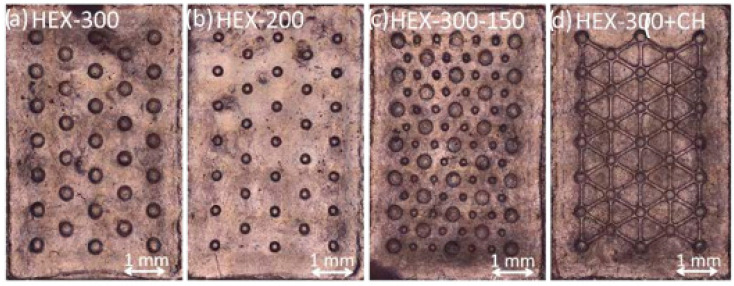
UHMWPE sample: (**a**) HEX-300; (**b**) HEX-200; (**c**) HEX-300-150; (**d**) HEX-300+CH.

**Figure 12 micromachines-14-00523-f012:**
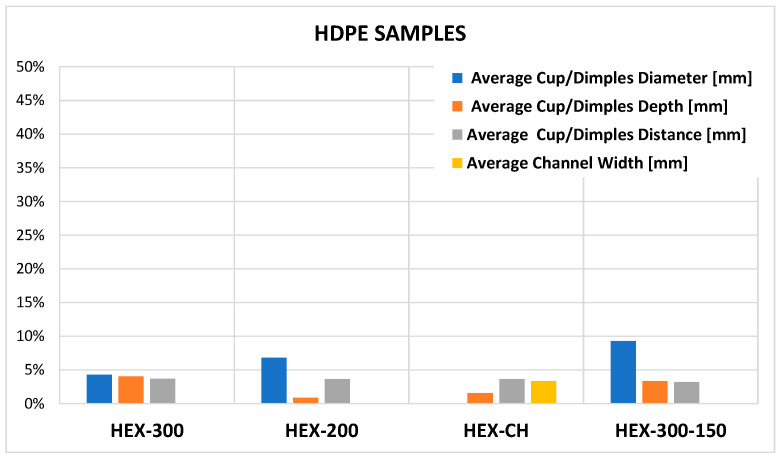
Percentage errors for HDPE samples.

**Figure 13 micromachines-14-00523-f013:**
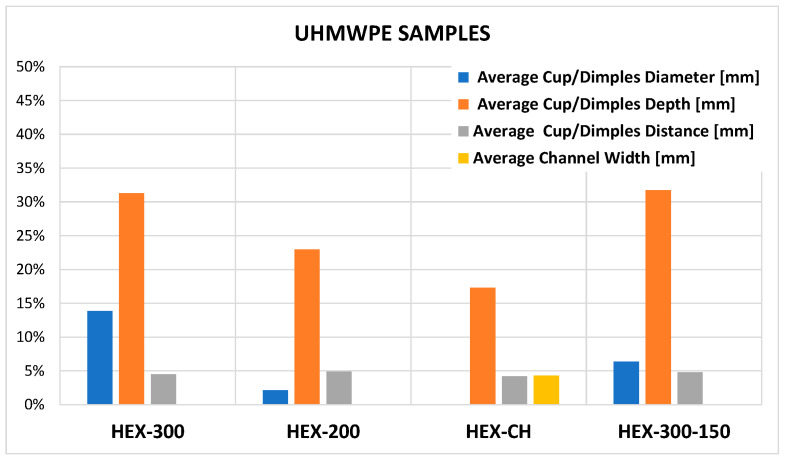
Percentage errors for UHMWPE samples.

**Figure 14 micromachines-14-00523-f014:**
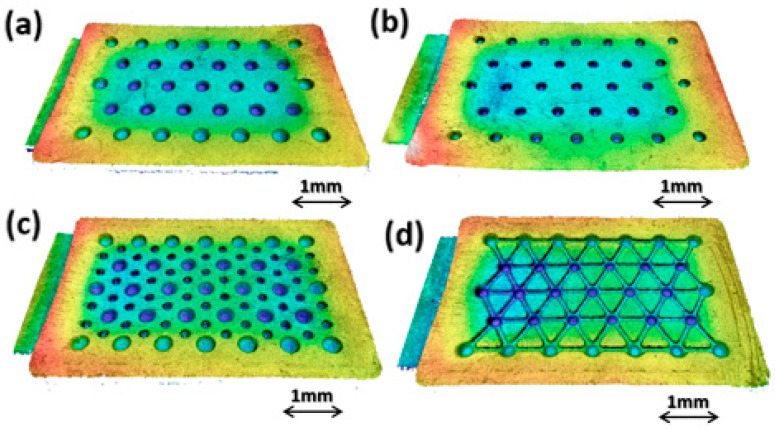
Isometric 3D view of the molded HDPE samples: HEX-300 (**a**); HEX-200 (**b**); HEX-300-150 (**c**); HEX-300+CH (**d**).

**Figure 15 micromachines-14-00523-f015:**
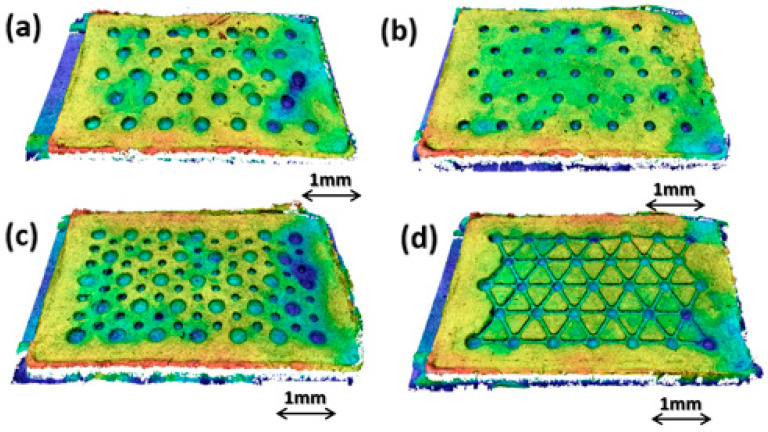
Isometric 3D view of the molded UHMWPE samples: HEX-300 (**a**); HEX-200 (**b**); HEX-300-150 (**c**); HEX-300+CH (**d**).

**Table 1 micromachines-14-00523-t001:** Micro-textures, geometrical parameters, and AF.

Micro-texture	Geometrical Parameters (μm)	AF = Cavity Area/Cell Area	Cavity Area Fraction AF%
HEX200	*D* = 200; *P* = 700	$\frac{\pi \cdot D^{2}}{2\cdot W\cdot L}$	7.4%
HEX300	*D* = 300; *P* = 700	$\frac{\pi \cdot D^{2}}{2\cdot W\cdot L}$	16.7%
HEX300- 150	*D*_1_ = 300; *D*_2_ = 150; *P*_1_ = 700; *P*_2_ = 409.74	$\frac{\pi \cdot \left(D_{1}^{2}+2\cdot D_{2}^{2}\right)}{2\cdot W\cdot L}$	25.0%
HEX300+CH	*D* = 300; *P* = 700; *CH* = 10; *CW* = 50; *CL* = 404.20	$\frac{24\cdot CW\cdot CL+2\pi \cdot D^{2}}{4\cdot W\cdot L}$	30.9%

*D* diameter; *P* pitch between features; *CW* channel width; *CL* channel length; *CH* channel height; *L* pattern cell length; *W* pattern cell width.

**Table 2 micromachines-14-00523-t002:** Comparison of main materials’ properties.

Property	Test Method	UHMWPE GUR^®^5113	Generic UHMWPE	HDPE Versalis Eraclene MP90C
Melting temperature (°C)	ISO 3146-C	130–135	130–135	137
Molecular weight (10^6^ g/mol)	ASTM D4001-20	3.7	3.5–7.5	0.05–0.25
Modulus of Elasticity (GPa)	ISO 527	0.8	0.5–0.8	0.4–4.0
Ultimate Tensile Strength (MPa)	ISO 527	37	39–48	17
Degree of crystallinity (%)	DSC ^1^	-	39–75	60–80
Wear rate (mm^3^/10^6^ cycles)	[23]	-	80–100	380–400

^1^ Differential Scanning Calorimetry.

**Table 3 micromachines-14-00523-t003:** Micro-EDM process parameters.

Parameter	Units	HEX-300 HEX-200	HEX-300-150 HEX-CH
Pulse width	(μs)	4	4
Pulse frequency	(Hz)	160	160
Current (Index)	-	100	100
Voltage	(V)	100	100
Gain	-	220	220
Gap	-	72	72
Energy (Index)	-	100	100
Regulation (Index)	-	4000	4000
Incremental depth	(μm)	0.9	1.2
Tool diameter	(mm)	0.41	0.16

**Table 4 micromachines-14-00523-t004:** Micro-EDM performance.

Parameter	Unit	HEX-300	HEX-200	HEX-CH	HEX-300-150
Depth error	(mm)	0.001	0.0006	0.0018	0.0007
Machining time	(s)	5965	6067	5600	5697
Average Speed	(mm/s)	0.62	0.62	0.60	0.63
MRR	(mm^3^/min)	0.0086	0.0086	0.0043	0.0045
TWR	-	0.18	0.20	1.56	1.21

**Table 5 micromachines-14-00523-t005:** Mold features’ measurements.

MOLD	HEX-300	HEX-200	HEX-300-150		HEX-CH
**Domes diameter**					
Nominal (mm)	0.300	0.200	0.300	0.150	-
Average (mm)	0.307	0.210	0.291	0.147	-
Mean Deviation (error) (mm)	+0.007	+0.010	−0.009	−0.003	-
std dev (mm)	0.004	0.004	0.007	0.005	-
**D** **omes height**					
Nominal (mm)	0.020	0.020	0.020		0.020
Average (mm)	0.023	0.022	0.022		0.024
Mean Deviation (error) (mm)	+0.003	+0.002	+0.002		+0.004
std dev (mm]	0.0005	0.0006	0.0009		0.0004
**Domes distance**					
Nominal (mm)	0.700	0.700	0.700	0.404	0.700
Average (mm)	0.698	0.698	0.698	0.403	0.699
Mean Deviation (error) (mm)	−0.002	−0.002	−0.002	−0.001	−0.001
std dev (mm]	0.003	0.002	0.003	0.002	0.003
**Channel Width**					
Nominal (mm)	-	-	-	-	0.050
Average (mm)	-	-	-	-	0.054
Mean Deviation (error) (mm)					+0.004
std dev )mm)	-	-	-	-	0.0026
**Surface Roughness Sa (µm)**	0.3				

**Table 6 micromachines-14-00523-t006:** Micro-injection molding process parameters.

Parameter	Unit	HDPE	UHMWPE
		Versalis Eraclene MP90C	Celanese GUR^®^5113
Melt Temperature	(°C)	200	260
Mold Temperature	(°C)	80	100
Injection Speed	(mm/s)	100	230
Holding Pressure	(MPa)	80	150
Holding Time	(s)	2	5
Cooling Time	(s)	5	10

**Table 7 micromachines-14-00523-t007:** HDPE sample features measurements.

HDPE Samples	HEX-300	HEX-200	HEX-300-150		HEX-CH
**Cups Diameter**					-
Nominal (mm)	0.300	0.200	0.300	0.150	-
Mold feature (mm)	0.307	0.210	0.291	0.147	-
Sample Average (mm)	0.294	0.224	0.301	0.169	-
Mean Deviation(error) (mm)	−0.013	0.014	0.01	0.022	-
std dev (mm)	0.029	0.025	0.010	0.007	-
**Cups Depth**					
Nominal (mm)	0.020	0.020	0.020		0.020
Mold feature (mm)	0.023	0.022	0.022		0.024
Sample Average (mm)	0.024	0.022	0.023		0.024
Mean Deviation (error) (mm)	0.001	0.000	0.001		0.000
std dev (mm)	0.0024	0.0014	0.0014		0.0018
**Cups Distance**					
Nominal (mm)	0.700	0.700	0.700	0.404	0.700
Mold feature (mm)	0.698	0.698	0.698	0.403	0.699
Sample Average (mm)	0.672	0.673	0.675	0.390	0.673
Mean Deviation (error) (mm)	−0.028	−0.027	−0.025	−0.014	−0.027
std dev (mm)	0.007	0.009	0.003	0.002	0.004
**Channels Width**					
Nominal (mm)	-	-	-	-	0.050
Mold feature (mm)	-	-	-	-	0.054
Sample Average (mm)	-	-	-	-	0.052
Mean Deviation (error) (mm)	-	-	-	-	−0.002
std dev (mm)	-	-	-	-	0.0012
**Surface Roughness Sa (µm)**	0.3				

**Table 8 micromachines-14-00523-t008:** UHMWPE sample features’ measurement.

UHMWPE Samples	HEX-300	HEX-200	HEX-300-150		HEX-CH
**Cups Diameter**					
Nominal (mm)	0.300	0.200	0.300	0.150	-
Mold feature (mm)	0.307	0.210	0.291	0.147	-
Sample Average (mm)	0.265	0.205	0.265	0.141	-
Mean Deviation (error) (mm)	−0.042	−0.005	−0.026	−0.006	-
std dev (mm)	0.017	0.004	0.017	0.005	-
**Cups Depth**					
Nominal (mm)	0.020	0.020	0.020		0.020
Mold feature (mm)	0.023	0.022	0.022		0.024
Sample Average (mm)	0.03	0.028	0.029		0.028
Mean Deviation (error) (mm)	0.007	0.006	0.007		0.004
std dev (mm)	0.0019	0.0015	0.0028		0.0006
**Cups Distance**					
Nominal (mm)	0.700	0.700	0.700	0.404	0.700
Mold feature (mm)	0.698	0.698	0.698	0.403	0.699
Sample Average (mm)	0.667	0.664	0.667	0.382	0.669
Mean Deviation (error) (mm)	−0.031	−0.034	−0.031	−0.021	−0.030
std dev (mm)	0.017	0.006	0.017	0.006	0.011
**Channels Widt** **h**					
Nominal (mm)	-	-	-	-	0.050
Mold feature (mm)	-	-	-	-	0.054
Sample Average (mm)	-	-	-	-	0.051
Mean Deviation (error) (mm)	-	-	-	-	−0.003
std dev (mm)	-	-	-	-	0.0030
**Surface Roughness Sa (µm)**	0.2				

## Data Availability

No new data were created, or data are unavailable due to privacy or ethical restrictions.

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
