# Peer review of "Replication Study of Molded Micro-Textured Samples Made of Ultra-High Molecular Weight Polyethylene for Medical Applications"

_micromachines, 2023, doi:10.3390/mi14030523_

Round 1
Reviewer 1 Report
This paper is very well written and describe in detail a manufacturing experiment on the replication of Micro-textures on Ultra-High Molecular Weight Polyethylene using Injection Moulding.
Here are some suggestions for improvements.
The novelty of the paper, although visible in places, could be highlighted better in the introduction and in the conclusion section. Showing more clearly the gaps in knowledge and then the findings.
At the end of section 1, the authors state that “In this work, the micro-texturing surface replication capability of UHMWPE via injection molding is investigated”. This suggest that the main aim is the assessment of such capability (rather than specifically replicating the provided textures) but the rationale behind the selected Micro-textures design described in section 2 is not clear. The design is described in terms of the end application but the authors should also explain why those textures were selected and why they are relevant for the replication capability investigation mentioned.
In the conclusion, the final statement is a bit vague” The experimental results and 234 samples inspection show that the UHMWPE material can be processed via injection molding and it is able to fill micro-cavities even if the process needs further optimization.” More specifics could be added.
Author Response
Rebuttal Letter to the reviewers of the paper entitled
“Replication study of molded micro-textured samples made of Ultra-High Molecular Weight Polyethylene for medical applications”
Micromachines MDPI
Special Issue WCMNM 2022
Dear Reviewers,
First, the authors want to thank you for contributing to improving the manuscript.
All comments were addressed, taking into account the reviewers’ advices. Here is a description of how the authors revised the paper by responding to each reviewer's comment.
Review 1
This paper is very well written and describes in detail a manufacturing experiment on the replication of Micro-textures on Ultra-High Molecular Weight Polyethylene using Injection Moulding.
Here are some suggestions for improvements.
The novelty of the paper, although visible in places, could be highlighted better in the introduction and in the conclusion section. Showing more clearly the gaps in knowledge and then the findings.
Authors’ response:
The authors thank the reviewer for his/her comment because it contributes at improving the quality and clearness of the paper. Abstract, introduction, and conclusion sections have been widely revised and improved according to the reviewer’s comment. The abstract has been clarified and the main quantitative results are now reported.
The introduction section has been strongly rearranged and enhanced with more information about the scientific context, open issues, and challenges. With this purpose, the current state-of-the-art about surface micro-texturing strategy for enhanced tribological properties has been presented in detail and also quantitatively by reporting the target coefficient of friction (COF=0.01-0.02) and the application wear rates (0.5-1 mm/y). Furthermore, after a more detailed description of the UHMWPE material, a comprehensive description of the manufacturing issues and challenges of this material has been presented. Previous works demonstrate that the injection molding of this material is still a challenge.
The replication capability of surface micro-texturing by micro-injection molding of the UHMWPE is clearly reported and compared with a conventional HDPE. The work contributes to the definition of the processability and performance of this important material, which is characterized by an essentially zero melt flow index, poor dissolution, extremely high melt viscosity and it is hard to be processed by conventional extrusion and injection molding technologies. These issues are particularly important at the micro-scale, where poor scientific contributions are present in literature.
The work addresses the lack of knowledge in the replication capability of micro-features by micro-injection molding of a “difficult-to-process” material such as UHMWPE. The relevance of this material and the impact of its surface micro-texturing in improving the tribological performance in many applications are the main motivations of this work. Micro-injection molding of such material is very challenging due to the material chemical and rheological properties. High melt viscosity, very low melt flow index and poor dissolution of UHMWPE hinder its processability and replication capability at the micro-scale. Therefore, the authors are confident that this work will contribute to the development.
In order to clarify the research, two new figures, 1 and 5, have been added, thus showing the research workflow and the materials and methods, respectively. Furthermore, all tables reporting the measurements, tables 5, 7 and 8, have been improved and results have been discussed in more detail at the end of section 5.
Revised parts are marked in yellow in the new version of the paper.
At the end of section 1, the authors state that “In this work, the micro-texturing surface replication capability of UHMWPE via injection molding is investigated”. This suggest that the main aim is the assessment of such capability (rather than specifically replicating the provided textures) but the rationale behind the selected Micro-textures design described in section 2 is not clear. The design is described in terms of the end application but the authors should also explain why those textures were selected and why they are relevant for the replication capability investigation mentioned.
Authors’ response:
The authors agree with the reviewer’s comment. The main aim of the work was not clearly explained in the first version of the manuscript. For this reason, authors widely revised all the manuscript and particularly the introduction section, trying to better support the rationale behind. Section 1 now reports a clear explanation of the aim, open issues and challenges of the technology, with reference to the application. Leveraging the new description of the state-of-the-art presented in section 1, the design description has been improved in the manuscript (highlighted in yellow) and also with new figures (1 and 5).
In the conclusion, the final statement is a bit vague” The experimental results and samples inspection show that the UHMWPE material can be processed via injection molding and it is able to fill micro-cavities even if the process needs further optimization.” More specifics could be added.
Authors’ response:
The authors agree with the reviewer about the missing of clearness of conclusions, therefore, the conclusions have been revised, also accordingly to other reviewer’s comments. The results have been discussed in more detail in section 5. Here follows the revised conclusions section, with more detail about the results:
“The present study investigates UHMWPE's micro-texturing surface replication capability via injection molding. Four different micro-textures cavities have been designed and fabricated on a steel mold by micro-EDM milling and used to test the micro-injection process replication. Two polymers, HDPE and UHMWPE, have been processed via micro-injection molding, comparing the replication results. The analysis shows that HDPE samples present percentage errors that span from 1% to 9% resulting in an average error of 4.3%. In comparison, the UHMWPE samples display percentage errors ranging from 2% to 31%, with an average error of 11.4%. Higher errors occur on the micro-feature depth with a nominal value of 20mm, in which an error of a few units produces a high value of percentage deviation. On the contrary, the diameters are well replicated for all the samples where the maximum error is under 10%.
The experimental results confirm that UHMWPE samples can be successfully fabricated by micro-injection molding, although the replication capability of micro-features and micro-textures is still challenging due to the rheological properties of the material. High-resolution microscopy reveals a lower quality in the filling of UHMWPE samples compared with the HDPE, thus proving that the process can be further optimized”.

Reviewer 2 Report
1. The abstract should be broadened to give additional quantitative results.
2. The present abstract was insufficient, please include the abstract's "take-home" message.
3. Rearrange keywords alphabetically.
4. The Reviewer do not see the novel in the present article. My examination revealed that several similar previous publications appear to appropriately address the issues you have brought up in the current submission. Please emphasize it more advance in the introduction section if there are any more truly something really new.
5. Previous study related needs to explain in the introduction section consisting of their work, their novelty, and their limitations to show the research gaps that intend to be filled in the present study.
6. In the introduction section, please explain the beneficial of textured surface in application of implant, such as lowering coefficient of friction, improve hydrodynamic pressure, and others. It needs to explain at least one paragraph. Relevan reference suggested to incorporated and discussed as follows: The Effect of Bottom Profile Dimples on the Femoral Head on Wear in Metal-on-Metal Total Hip Arthroplasty. J. Funct. Biomater. 2021, 12, 38. https://doi.org/10.3390/jfb12020038
7. To improve the reader's understanding of the materials and methods section simpler for them to grasp rather than only relying on the predominate text as it currently exists, the authors could incorporate figures that illustrate the workflow of the current study.
8. Other information about the tool, such as the manufacturer, country, and specifications, should be provided.
9. Important information that must be mentioned in the publication relates to the error and tolerance of the experimental equipment utilized in this investigation. As a result of the disparate findings in subsequent research by other researchers, it would be a useful discussion.
10. Results must be compared to similar past research.
Author Response
Rebuttal Letter to the reviewers of the paper entitled
“Replication study of molded micro-textured samples made of Ultra-High Molecular Weight Polyethylene for medical applications”
Micromachines MDPI
Special Issue WCMNM 2022
Dear Reviewers,
First, the authors want to thank you for contributing to improving the manuscript.
All comments were addressed, taking into account the reviewers’ advices. Here is a description of how the authors revised the paper by responding to each reviewer's comment.
Review 2
- The abstract should be broadened to give additional quantitative results.
Authors’ response:
The authors agree with the reviewer’s comment. The abstract has been widely improved according to the reviewer’s advices. Quantitative results have been clearly reported, thus showing the contribution of the paper. Revised parts are marked in yellow in the new version of the paper.
- The present abstract was insufficient, please include the abstract's "take-home" message.
Authors’ response:
As abovementioned in the previous comment response, the abstract now reports the “take-home” message. The replication capability of surface micro-texturing by micro-injection molding of the UHMWPE is clearly reported and compared with a conventional HDPE. The work contributes to the definition of the processability and performance of this important material, which is characterized by an essentially zero melt flow index, poor dissolution, extremely high melt viscosity and it is hard to be processed by conventional extrusion and injection molding technologies. These issues are particularly important at the micro-scale, where poor scientific contributions are present in literature.
- Rearrange keywords alphabetically.
Authors’ response:
Keywords have been rearranged in alphabetical order.
- The Reviewer do not see the novel in the present article. My examination revealed that several similar previous publications appear to appropriately address the issues you have brought up in the current submission. Please emphasize it more advance in the introduction section if there are any more truly something really new.
- Previous study related needs to explain in the introduction section consisting of their work, their novelty, and their limitations to show the research gaps that intend to be filled in the present study.
Authors’ response to reviewer’s comments 4 and 5:
The introduction section has been broadened and widely re-structured in order to bring out the scientific challenge, the aim, previous contributions and/or lacks of scientific contributes on this topic. Authors tried to present more clearly the issues expanding the description of state of the art related to both the final application target and the manufacturing issues. The revised introduction section is more clear and brings the reader to recognize the current lack of knowledge and the contribution of this paper to fill this gap.
The work addresses the lack of knowledge in replication capability of micro-features by micro-injection molding of a “difficult-to-process” material such as UHMWPE. The relevance of this material and the impact of its surface micro-texturing in improving tribological performance in many applications are the main motivations of this work. Micro-injection molding of such material is very challenging due to the material chemical and rheological properties. High melt viscosity, very low melt flow index, and poor dissolution of UHMWPE hinder its processability and replication capability at the micro-scale. Therefore, authors are confident that this work will contribute to the development.
- In the introduction section, please explain the beneficial of textured surface in application of implant, such as lowering coefficient of friction, improve hydrodynamic pressure, and others. It needs to explain at least one paragraph. Relevant reference suggested to incorporated and discussed as follows: The Effect of Bottom Profile Dimples on the Femoral Head on Wear in Metal-on-Metal Total Hip Arthroplasty. J. Funct. Biomater. 2021, 12, 38. https://doi.org/10.3390/jfb12020038
Authors’ response:
The authors thank the reviewer for this comment because it contributes to the improvement of the paper. According to the reviewer’s comment, the introduction section has been widely improved. The authors agree with the reviewer about a lack of clearness of the benefits of surface micro-texturing. The revised introduction now contains a description of state of the art about the surface micro-texturing as a technical strategy to improve the tribological properties and particularly by lowering the coefficient of friction (COF) and, the wear rate. This description is not limited to qualitative consideration, but it reports reference values of the COF (0.02) and target wear rate for the specific material and application (i.e., 0.5-1 mm/y).
Furthermore, authors recognized a lack of clarity in the description of the challenge of the processability and replication capability of the UHMWPE. Therefore, the introduction was enhanced with a brief description of the state-of-the-art about manufacturing issues of this material at micro-scale by injection molding. Now the reader is more correctly guided, step-by-step, to recognize the contribution of the work.
The authors thank the reviewer for the suggested paper. It is very interesting because it describes very well the final application and experimental tests. The suggested paper has been studied and cited in the introduction section.
- To improve the reader's understanding of the materials and methods section simpler for them to grasp rather than only relying on the predominate text as it currently exists, the authors could incorporate figures that illustrate the workflow of the current study.
Authors’ response:
The reviewer’s comment is very useful in improving the quality of the paper. Two new figures, 1 and 5, have been developed and included into the paper. The work flow is described by the flow chart reported in figure 1, while the “materials and methods” are clearly depicted in figure 5. These two figures help the reader in fixing in mind the main steps and method of the work.
- Other information about the tool, such as the manufacturer, country, and specifications, should be provided.
Authors’ response:
The machines and tools descriptions have been improved according to the reviewer’s comment.
Important information that must be mentioned in the publication relates to the error and tolerance of the experimental equipment utilized in this investigation. As a result of the disparate findings in subsequent research by other researchers, it would be a useful discussion.
Authors’ response:
The authors agree with the reviewer. The metrological characterization was performed by means of a confocal microscope Zeiss. The manuscript now reports not only information about the manufacturer but also a estimate of the measurement accuracy of the equipment. Hereafter, the description of the measurement equipment is reported:
“Profiles and surface roughness of inserts and samples were acquired via confocal microscope Zeiss Axio CSM 700 (Carl Zeiss Microimaging GmbH, Jena, Germany). The acquired images have a spatial resolution Rs of 1.824 mm/pixel that allows for a feature resolution Rf of 5.5 mm considering 3 pixels spanning the minimum size feature and a measurement resolution Rm of 0.1824 mm”.
- Results must be compared to similar past research.
Authors’ response:
The authors thank the reviewer for his comment because they recognize that it is a gap to be filled. In order to improve the results discussion and their relevance compared to other similar past researches, preliminarily, two main actions have been applied in the revision process: 1) enhance the clearness of results by improving tables 5, 7, and 8 and figures 11 and 12; 2) Results have been more accurately and quantitatively discussed in section 5.
The authors performed extensive literature state of the art to improve their knowledge of similar past research. Currently, the injection molding of UHMWPE for micro-structured surfaces is still a challenging task, as reported in the paper’s introduction. In fact, due to its physical characteristic (in particular, the melting point), the material is usually formed and structured by different methods such as laser surface texturing, lithography, micro-machining, hot embossing, hot stamping, and electrical discharge [Rahaman 2022, Quintanilla-Correa 2021, Hussain 2020]. Regarding injection molding, the references reported in the present paper [14-18] are related to manufacturing test samples showing well-known shapes such as dogbone, plates, etc., and not to micro-textured surfaces.
Thus, presenting a detailed comparison with similar past papers is not easy. In a recent article (2020), Liang et al. fabricated UHMWPE microplastic parts with a replicated micro-groove array by micro ultrasonic powder molding. The groove arrays have the following dimensions: a bottom radius of 70 μm and depth of 110 μm that are bigger than those of the channel HEX-CH (width of 50 μm and depth of 20 μm). Still, the obtained results are comparable in terms of replicability. The replication rates obtained by Liang are in the range of 94-98%, while in the present paper, the replication errors for channels are in the range of 4-5%. Despite this, we chose not to display this comparison because the textured parts are obtained with different processes, which enforces the present research's novelty.
M.D.M. Rahman, Md.A.S. Biswas, K.N. Hoque, Recent development on micro-texturing of UHMWPE surfaces for orthopedic bearings: A review, Biotribology, Volume 31, 2022, 100216, ISSN 2352-5738, https://doi.org/10.1016/j.biotri.2022.100216.
D.I. Quintanilla-Correa, L. Pena-Paras, D. Madonaldo-Cortes, M.C. Rodriguez-Villalobos, M.A.L. Hernandez-Rodrigues, State of the art of surface texturing for biotribology applications, Int Journal of Modern Manufacturing Technol, ISSN2067–3604,Vol.XIII,No.3/2021, https://doi.org/10.54684/ijmmt.2021.13.3.143
Hussain M.; Naqvi R.A.; Abbas N.; Khan S.M.; Nawaz S.; Hussain A.; Zahra N.; Khalid M.W.; Ultra-High-Molecular-Weight-Polyethylene (UHMWPE) as a Promising Polymer Material for Biomedical Applications: A Concise Review. Polymers 2020, 12, 323. https://doi.org/10.3390/polym12020323
Liang, Y.j. Liu, S.g. Chen, J. Ma, X.y. Wu, H.y. Shi, L.y. Fu, B. Xu, Fabrication of microplastic parts with a hydrophobic surface by micro ultrasonic powder moulding, Journal of Manufacturing Processes, Volume 56, Part A, 2020, Pages 180-188, ISSN 1526-6125, https://doi.org/10.1016/j.jmapro.2020.04.086.

Round 2
Reviewer 2 Report
Further comments in authors revised manuscript is given:
1. In authors revised manuscript, new added paragraph is given in line 35-41. It explain related to arthroplasty. As noted, it surgeries consist of total hip prosthesis and hip resurfacing. Where, survece texturing is given mostly in total hip prosthesis. Authors needs to incorporated new information in revised manuscript. Supporting reference as follows: Adopted Walking Condition for Computational Simulation Approach on Bearing of Hip Joint Prosthesis: Review over the Past 30 Years. Heliyon 2022, 8, e12050. https://doi.org/10.1016/j.heliyon.2022.e12050
2. Overall, discussion in the present article is extremely poor. The Authors must extend their discussion and make a comprehensive explanation. Just not simply mention the results with brief explanation. Scpecific contecxt on biomechanics, biotribology, and biomaterials should be explained. For highlight its field, the authtors would refer from the other previous work of Ammarullah et al. group, please see recommended reference in comment number 1.
3. In line 42-52, the aurthors has been explained the explanation of choosing materials in gip implant, but it neef to more advanced. For example, discuss each of materials advantages and disadcantages. Such as titanium mentioned in line 42 is good in corrosion and biocompability aspect.
4. Before moving on to the conclusion section, the present study's limitation must be added at end of the discussion section.
5. Further research should indeed be mentioned in the conclusion section.
6. The reference needs to be enriched from the literature published five years back. MDPI reference is strongly recommended.
7. The manuscript needs to be proofread by the authors since it has grammatical and language issues.
8. The authors need to provide a graphical abstract for submission after the revision stage.
Author Response
Rebuttal Letter to the reviewer of the paper entitled
“Replication study of molded micro-textured samples made of Ultra-High Molecular Weight Polyethylene for medical applications”
Micromachines MDPI
Special Issue WCMNM 2022
Dear Reviewer,
Thank you for your comments. The authors appreciate the effort and the contribution to improving the manuscript.
All issues were addressed, taking into account the reviewer’s comments. Here is a description of how the authors revised the paper. Two versions of the manuscript have been uploaded: a clean manuscript and a version with revised parts marked in yellow.
Reviewer 2
Further comments in authors revised manuscript is given:
- In authors revised manuscript, new added paragraph is given in line 35-41. It explain related to arthroplasty. As noted, it surgeries consist of total hip prosthesis and hip resurfacing. Where, survece texturing is given mostly in total hip prosthesis. Authors needs to incorporated new information in revised manuscript. Supporting reference as follows: Adopted Walking Condition for Computational Simulation Approach on Bearing of Hip Joint Prosthesis: Review over the Past 30 Years. Heliyon 2022, 8, e12050. https://doi.org/10.1016/j.heliyon.2022.e12050
Authors’ response:
The revised manuscript now addresses the required explanation related to arthroplasty. We included the suggested reference; see revised section 1 “Introduction”. The bearing couple components are now better identified, and figure 1 has been added for this purpose. See also the response to comment 2. Here are some new paragraphs of the revised paper:
In section 1 “Introduction”:
“The current service life of an artificial hip joint prosthesis is about 15 years and a research gap must be filled to improve the quality and life of these products [3]. Improvements are required in materials performance, biomechanics, biotribology, biomaterials, mechanical analysis methodology and physiology [3]”.
And:
“The current state-of-the-art on materials for hip prostheses is widely presented in a recent review paper [6]. Two procedures are adopted in hip joint surgery: total hip arthroplasty and hip resurfacing [3]. The former consists of replacing femur head, femoral stem, and acetabulum cup, while the latter consists of replacement of the bearing couple (femoral head and acetabular cup) at the interface between femoral stem. In hip resurfacing it is not required the replacement of femoral stem with implant component (figure 1).
In all joint implants, the bearing couple is the main functional component for load-bearing and movement articulation since it provides continuous contact and mechanical action transmission between structural components in patient's activity [3]. Bearing couples are subjected to friction, wear, and surface damage affecting overall performance and prosthesis lifetime. In last decade, several studies have been focused on bearing couple in order to increase the life of implants and minimize implant failures [3]”.
Figure 1. Total hip arthroplasty (a) and hip resurfacing (b) [3].
In section 2:
“A recent review over 30 years showed that computational simulation is an effective approach in assessing bearing, friction, wear, surface damages, performance, and failure in artificial joints [3]”.
[3] Jamari, J., et al. "Adopted walking condition for computational simulation approach on bearing of hip joint prosthesis: review over the past 30 years." Heliyon (2022): e12050.
- Overall, discussion in the present article is extremely poor. The Authors must extend their discussion and make a comprehensive explanation. Just not simply mention the results with brief explanation. Specific context on biomechanics, biotribology, and biomaterials should be explained. For highlight its field, the authtors would refer from the other previous work of Ammarullah et al. group, please see recommended reference in comment number 1.
Authors’ response:
The overall discussion in the article has been improved thanks to this reviewer’s comment. In order to improve the clarity and completeness of the explanations, several new paragraphs have been added. Furthermore, a new section entitled “Surface micro-texturing for enhanced tribological properties in bearing couples” has been included. The discussion is now explained and supported by further references, as suggested by the reviewer. According to the reviewer’s comment, we also referred to the valuable previous works of Dr. Ammarullah's research group.
Discussion extension in the section 1 “Introduction”:
“The current state-of-the-art on materials for hip prostheses is widely presented in a recent review paper [6]. Two procedures are adopted in hip joint surgery: total hip arthroplasty and hip resurfacing [3]. The former consists of replacing femur head, femoral stem, and acetabulum cup, while the latter consists of replacement of the bearing couple (femoral head and acetabular cup) at the interface between femoral stem. In hip resurfacing it is not required the replacement of femoral stem with implant component (figure 1).
In all joint implants, the bearing couple is the main functional component for load-bearing and movement articulation since it provides continuous contact and mechanical action transmission between structural components in patient's activity [3]. Bearing couples are subjected to friction, wear, and surface damage affecting overall performance and prosthesis lifetime. In last decade, several studies have been focused on bearing couple in order to increase the life of implants and minimize implant failures [3]”.
Figure 1. Total hip arthroplasty (a) and hip resurfacing (b) [3].
This figure reproduces figure 1 of the cited paper:
Jamari, J., et al. "Adopted walking condition for computational simulation approach on bearing of hip joint prosthesis: review over the past 30 years." Heliyon (2022): e12050. https://doi.org/10.1016/j.heliyon.2022.e12050
Published by Elsevier Open-Access Journal Heliyon. Since it is an open-access Journal, the copyright is held by the authors. Therefore, the authors of the cited work have been contacted by email to ask for authorization to reproduce their figure. The corresponding author of the cited paper, Dr. Muhammad Imam Ammarullah, kindly authorized the authors to use the figure in the submitted manuscript. See Annex 1.
Other new references have been included to support the discussion, especially in section 2.
- In line 42-52, the aurthors has been explained the explanation of choosing materials in gip implant, but it neef to more advanced. For example, discuss each of materials advantages and disadcantages. Such as titanium mentioned in line 42 is good in corrosion and biocompability aspect.
Authors’ response:
The manuscript has been revised according to the reviewer’s comment. The materials are cited, and the advantages and disadvantages have been explicitly mentioned. Some specific references about the bearing couple materials have been added. In addition, these aspects are addressed in a recent review [6]. Hereafter we report some sentences added to the revised manuscript:
“Titanium alloys are preferred as bearing components due to their mechanical performance, biocompatibility and corrosion resistance, but their debris dispersion can trigger osteolysis [1, 2]”.
And:
“The current service life of an artificial hip joint prosthesis is about 15 years and a research gap must be filled to improve the quality and life of these products. Improvements are required in materials performance, biomechanics, biotribology, biomaterials, mechanical analysis methodology, and physiology [3]”.
“The current state-of-the-art on materials for hip prostheses is widely presented in a recent review paper [6]”.
- Before moving on to the conclusion section, the present study's limitation must be added at end of the discussion section.
Authors’ response:
To address the reviewer’s comment, the manuscript has been modified, including the following paragraph, regarding the study’s limitations, in the final part of the discussion section.
“Thus, the experimental results verified that, differently from previous research, UHMWPE samples can be micro-molded by injection process. However, the replication capability of micro-features and micro-textures is still a challenging task due to the rheological properties of the material. It must also be underlined that the mold cavities are, on purpose, narrow and generally difficult to fill. The defects recorded on the samples during the experimentation could not appear with thicker cavities or cavities with higher aspect ratio, as demonstrated by the fact that defects occur far from the gate. The geometry has been designed with the goal of studying the limits of injection molding when UHMWPE is injected, providing a helpful indication for prosthesis mold design.”
Furthermore, these limitations are also summarized in other lines of this section as follows:
“Higher errors occur on the micro-feature depth with a nominal value of 20mm, in which an error of a few units produces a high value of percentage deviation.”
“As shown in figure 15, the molding quality of the UHMWPE samples degrades along the sample length. In particular, the molding quality and replication capability appear acceptable closer to the gate region (left) and for about two third of the sample length, while they degrade approaching the final part of the cavity (right).”
“Material viscosity, in conjunction with the cavity geometry, results in a pressure drop and a drastic cooling of the melt front, thus determining a low molding quality.”
- Further research should indeed be mentioned in the conclusion section.
Authors’ response:
Thanks to the reviewer for this comment that improves the conclusion section. The following sentence was added to the revised version of the paper.
“Further research will regard, as mentioned, the optimization of injection molding process parameters and, in particular, the accurate definition of injection pressure and temperature, whose setting is of paramount importance. In fact, the molding conditions at the polymer-mold interface play a fundamental role. Also, a new formulation of a medical-compliant injection moldable Celanese GUR showing improved viscosity is under processing and testing. Finally, further research will be focused on the tribology of the developed surface micro-texturing. An experimental campaign will be aimed at accurately measuring the coefficient of friction and wear rates in different operation conditions.”
- The reference needs to be enriched from the literature published five years back. MDPI reference is strongly recommended.
Authors’ response:
The authors thank the reviewer for this comment. To answer this request, the following studies, mostly from MDPI, in the range 2018-2023 were added to the revised version of the paper:
- Eliaz, N. Corrosion of Metallic Biomaterials: A Review. Materials2019, 12, 407. https://doi.org/10.3390/ma12030407
- Jamari, J., et al. "Adopted walking condition for computational simulation approach on bearing of hip joint prosthesis: review over the past 30 years." Heliyon (2022): e12050.
- Merola, M.; Affatato, S. Materials for Hip Prostheses: A Review of Wear and Loading Considerations. Materials2019, 12, 495. https://doi.org/10.3390/ma12030495
- Hussain M.; Naqvi R.A.; Abbas N.; Khan S.M.; Nawaz S.; Hussain A.; Zahra N.; Khalid M.W.; Ultra-High-Molecular-Weight-Polyethylene (UHMWPE) as a Promising Polymer Material for Biomedical Applications: A Concise Review. Polymers2020, 12, 323. https://doi.org/10.3390/polym12020323
- Yilmaz, G.; Ellingham, T.; Turng, L.S. Improved Processability and the Processing-Structure-Properties Relationship of Ultra-High Molecular Weight Polyethylene via Supercritical Nitrogen and Carbon Dioxide in Injection Molding. Polymers2018, 10, 36. https://doi.org/10.3390/polym10010036
- Lu, Ping, and Robert JK Wood. "Tribological performance of surface texturing in mechanical applications-A review." Surface Topography: Metrology and Properties (2020).
- Bracco, P.; Bellare, A.; Bistolfi, A.; Affatato, S. Ultra-High Molecular Weight Polyethylene: Influence of the Chemical, Physical and Mechanical Properties on the Wear Behavior. A Review. Materials 2017, 10, 791. https://doi.org/10.3390/ma10070791.
- The manuscript needs to be proofread by the authors since it has grammatical and language issues.
Authors’ response:
The manuscript has been proofread by the authors and by a third-party colleague. All issues seem to be solved. Thank you.
- The authors need to provide a graphical abstract for submission after the revision stage.
Authors’ response:
Thank you for the advice. A graphical abstract has been included in the submission.
ANNEX 1. AUTHORIZATION TO THE USE OF FIGURE 1
Dear Dr. Vito Basile,
Many thanks for your email and glad to know you, sir.
I am Muhammad Imam Ammarullah, and you can call me Imam. I am a lecturer at Department of Mechanical Engineering, Universitas Pasundan, Indonesia. Also, I am a program coordinator at Biomechanics and Biomedics Engineering Research Centre, Universitas Pasundan, Indonesia.
I am very happy to hear about your interest in our review article. Because our review article is published on Heliyon, the copyright is held by the author.
Through my response to your email, I am the corresponding author of the article: Adopted walking condition for computational simulation approach on bearing of hip joint prosthesis: review over the past 30 years (https://doi.org/10.1016/j.heliyon.2022 . e12050).
Giving formal permission for you to reproduce the figure in your manuscript. I hope that the permission can be useful for your planned manuscripts and can improve the overall quality of your work.
I am also very open if you are interested in my other research which you can access through the "Research" link in the footer of this email. And I am also very happy when you give the opportunity and research cooperation with you in the future.
If there is anything that I can provide to you, please let me know.
Thank you for your kind attention.
Best regards,
Muhammad Imam Ammarullah, M.Eng. (Hons)
Program Coordinator
Biomechanics and Biomedics Engineering Research Centre
Center of Excellence for Research
Pasundan University
Bandung 40153, Indonesia
Chairman
Tribology and Surface Engineering Laboratory
Department of Mechanical Engineering
Pasundan University
Bandung 40153, Indonesia
Social Media
SMS | Call | WhatsApp | Telegram: +62895335922435
Instagram | Twitter | Line | Tiktok: Imamammarullah
Facebook | LinkedIn: Muhammad Imam Ammarullah
E-mail: imamammarullah@gmail.com | imamammarullah@unpas.ac.id
Tumblr: imamammarullah.tumblr.com
Research
Google Scholar | Scopus | Web of Science | Research Gate
ORCID | Semantic Scholars | Academia | SciProfiles | Kudos
Pada tanggal Sel, 14 Feb 2023 pukul 22.50 Vito Basile <Vito.Basile@stiima.cnr.it> menulis:
Dear Dr. Muhammad Imam Ammarullah,
let me introduce myself. My name is Vito Basile, and I am a researcher at the Italian National Research Council.
As the corresponding author, I am submitting a paper and I will cite your recent valuable work:
Jamari, J., et al. "Adopted walking condition for computational simulation approach on bearing of hip joint prosthesis: review over the past 30 years." Heliyon (2022): e12050.
https://doi.org/10.1016/j.heliyon.2022.e12050
I will use your discussion to support my research, and I would like to use your figure 1.
I made a check, and I verified that the Heliyon Journal is an open-access journal.
With this email, I ask you and other authors for the formal authorization to use your figure 1 as follows:
I look forward to receiving your answer as soon as possible.
Best regards,
Vito Basile

Round 3
Reviewer 2 Report
Appreciate the authors effort for their effort in revising their manuscript. The present form was improved. However, the revision made by authors in this stage has some substance that needs to address. It is important to avoid flaws and improve quality.
1. It is not consistent with the authors, where mentioned in line 98 “coefficient of friction”, but in line 172 giving abbreviations for the phrase as CoF. If the authors want to use “CoF”, please initiate it in line 98, not in line 172. Or it would be okay not using “CoF”.
2. Related to comment number 1, the authors recommended explaining the obtained value of coefficient of friction. This value was obtained from an experimental setup, either pin-on-disc or hip joint simulator as explained by Jamari et al. Suggested literature needs to be incorporated, DOI: 10.3390/jfb13020064
3. The quality of Figure 2 and 5 is encouraged to enhances due to bias quality.
4. In line 31-38, the authors explain the role of arthroplasty without any supporting references. In discussing related to materials, design, and performance. Suggested reference is needed ado adopted, please refer, DOI: 10.3390/su142013413 and 10.3390/ma14247554
5. In line 39-42, I am suggesting to develop the paragraph. It is too short. For more consistence in context of hip joint. For application in bearing, titanium alloy has been proven its outstanding ability compared to other metals. Please refer the relevant reference, DOI: 10.3390/met12081241
6. In line 61, the authors mention “PE, PP, PEEK, PA66…”, please mention the stans of its abbreviations.
7. In line 55-57 it should improve. Only mentioning coefficient of friction with their characteristic? For information, polymers are used as damper-role in avoiding failure fixation systems on the artificial joints due to impact load. One of the discussions is valuable for improving the paragraph. Relevant reference for the discussion, DOI: 10.1016/j.matpr.2022.02.055
8. In line 67, “figure 1” should be capitalize as “Figure 1”.
9. Line 85, please revise “AMUs (atomic mass units)” to “atomic mass units (AMUs)”.
Author Response
Rebuttal Letter to the reviewer of the paper entitled
“Replication study of molded micro-textured samples made of Ultra-High Molecular Weight Polyethylene for medical applications”
Micromachines MDPI
Special Issue WCMNM 2022
Dear Reviewer,
Thank you for your new review.
All issues were addressed. Hereafter is a description of how the authors revised the paper. Two versions of the manuscript have been uploaded: a clean manuscript and a version with revised parts marked in yellow.
Reviewer 2
Appreciate the authors effort for their effort in revising their manuscript. The present form was improved. However, the revision made by authors in this stage has some substance that needs to address. It is important to avoid flaws and improve quality.
- It is not consistent with the authors, where mentioned in line 98 “coefficient of friction”, but in line 172 giving abbreviations for the phrase as CoF. If the authors want to use “CoF”, please initiate it in line 98, not in line 172. Or it would be okay not using “CoF”.
The reviewer’s comment is correct. The acronym “CoF” has been defined in row 56, and it has been used in this form all over the manuscript.
- Related to comment number 1, the authors recommended explaining the obtained value of coefficient of friction. This value was obtained from an experimental setup, either pin-on-disc or hip joint simulator as explained by Jamari et al. Suggested literature needs to be incorporated, DOI: 10.3390/jfb13020064
The value of the hydrodynamic coefficient of friction is known in the literature. This value has been experimentally measured in cited references [1] and [10]. The authors think that these references are enough to support the reported value.
- The quality of Figure 2 and 5 is encouraged to enhances due to bias quality.
Figures 2, 3, 4, and 5 have been improved in quality.
- In line 31-38, the authors explain the role of arthroplasty without any supporting references. In discussing related to materials, design, and performance. Suggested reference is needed ado adopted, please refer, DOI: 10.3390/su142013413 and 10.3390/ma14247554
- In line 39-42, I am suggesting to develop the paragraph. It is too short. For more consistence in context of hip joint. For application in bearing, titanium alloy has been proven its outstanding ability compared to other metals. Please refer the relevant reference, DOI: 10.3390/met12081241
Authors’ response to the reviewer’s comments 4 and 5.
We thank the reviewer for the comment encouraging us to address more deeply the topics of arthroplasty and titanium alloy components in hip joint prostheses, but we think that this is not the papers’s main topic that investigates replication capability via injection molding of polymeric micro-textured surfaces. Furthermore, we consider that the references already cited in the paper [1-3, 6, 13] are exhaustive of the problem; hence, further discussion on these topics would confuse the reader on the manuscript aim.
- In line 61, the authors mention “PE, PP, PEEK, PA66…”, please mention the stans of its abbreviations.
All materials have been defined correctly according to the reviewer’s comment:
“Among biomaterials, Ultra High Molecular Weight Polyethylene (UHMWPE) has gained widespread application due to its superior properties [4, 5] compared with other materials i.e., Polyethylene (PE), Polypropylene (PP), Polyether Ether Ketone (PEEK), Polyamide PA66, zirconia-on-PE, alumina ceramic [1, 4]”.
- In line 55-57 it should improve. Only mentioning coefficient of friction with their characteristic? For information, polymers are used as damper-role in avoiding failure fixation systems on the artificial joints due to impact load. One of the discussions is valuable for improving the paragraph. Relevant reference for the discussion, DOI: 10.1016/j.matpr.2022.02.055
The authors agree with the reviewer. According to the reviewer’s comment, the damping function of the impact loads has been added at lines 55-57 of the manuscript.
- In line 67, “figure 1” should be capitalize as “Figure 1”.
The typo has been corrected.
- Line 85, please revise “AMUs (atomic mass units)” to “atomic mass units (AMUs)”.
The revision has been applied to the manuscript:
“As specified by ASTM [9], it has an exceptionally high molecular weight, greater than 3.1 million atomic mass units (AMUs)”.
